

# Delimiting the boundaries of sesamoid identities under the network theory framework

Gabriela Fontanarrosa[1,*], Jessica Fratani[2,*] and Miriam C. Vera[3]

[1] Instituto de Biodiversidad Neotropical (IBN), CONICET-UNT, Yerba Buena, Tucumán, Argentina
[2] Unidad Ejecutora Lillo (UEL), CONICET-Fundación Miguel Lillo, San Miguel, Tucumán, Argentina
[3] Laboratorio de Genética Evolutiva, Instituto de Biología Subtropical (IBS), CONICET-UNaM, Posadas, Misiones, Argentina
[*] These authors contributed equally to this work.

## ABSTRACT

Sesamoid identity has long been the focus of debate, and how they are linked to other elements of the skeleton has often been considered relevant to their definition. A driving hypothesis of our work was that sesamoids' nature relies deeply on their connections, and thus we propose an explicit network framework to investigate this subject in *Leptodactylus latinasus* (Anura: Leptodactylidae). Through the dissection of *L. latinasus'* skeleton, we modeled its anatomical network where skeletal elements were considered nodes while joints, muscles, tendons, and aponeurosis were considered links. The skeletal elements were categorized into canonical skeletal pieces, embedded sesamoids, and glide sesamoids. We inquired about the general network characterization and we have explored further into sesamoid connectivity behavior. We found that the network is structured in a modular hierarchical organization, with five modules on the first level and two modules on the second one. The modules reflect a functional, rather than a topological proximity clustering of the skeleton. The 25 sesamoid pieces are members of four of the first-level modules. Node parameters (centrality indicators) showed that: (i) sesamoids are, in general terms, peripheral elements of the skeleton, loosely connected to the canonical bone structures; (ii) embedded sesamoids are not significantly distinguishable from canonical skeletal elements; and (iii) glide sesamoids exhibit the lowest centrality values and strongly differ from both canonical skeletal elements and embedded sesamoids. The loose connectivity pattern of sesamoids, especially glides, could be related to their evolvability, which in turn seems to be reflected in their morphological variation and facultative expression. Based on the connectivity differences among skeletal categories found in our study, an open question remains: can embedded and glide sesamoids be defined under the same criteria? This study presents a new approach to the study of sesamoid identity and to the knowledge of their morphological evolution.

Corresponding author
Gabriela Fontanarrosa,
gab.fontanarrosa@gmail.com

## INTRODUCTION

Sesamoids have been intriguing scientists since the beginning of the last century (*Abdala et al., 2019*). No clear consensus exists regarding their identity nor their membership to the canonical skeleton. Sesamoids definitions, often, rely not only on their intrinsic properties but also have strong references to their connections to surrounding tissues (*Didomenico et al., 2014*; *Regnault, Hutchinson & Jones, 2016*). The following definitions serve as examples: "*Sesamoids are nodules of cartilage or bone formed in tendons or ligaments, especially where a tendon passes over an angulation of the skeleton*" (*Hall, 2005*); "*Sesamoids are periarticular skeletal elements, which initially form in juxtaposition to or independently of bones and joints. They are commonly related to tendons and ligaments (...)*" (*Abdala et al., 2019*). Moreover, beyond those definitions, *Jerez, Mangione & Abdala (2010)* categorized sesamoids into four types, also based on their specific relationship to surrounding tissues: (i) embedded sesamoid (ES) (surrounded in all their surfaces by tendinous tissue); (ii) inter-osseous sesamoid (loosely attached to the closest ligaments); (iii) glide sesamoid (GS) (associated to tendons, but not surrounded by them and not fixed to them); and (iv) supporting sesamoid (serving as muscle attachment areas to the corresponding bones).

Since these definitions and classifications commonly refer to sesamoid connections, researchers were considering sesamoids, in an implicit but pervasive way, within a network framework. Network analyses have been increasingly used in the context of comparative vertebrate morphology answering questions about modularity and evolvability, among others (*Esteve-Altava et al., 2013*; *Rasskin-Gutman & Esteve-Altava, 2014*; *Dos Santos et al., 2017*; *Diogo et al., 2018*). Modular organization allows each structure to evolve semi-independently and promotes evolvability, avoiding deleterious pleiotropic effects (*Wagner & Zhang, 2011*; *Esteve-Altava et al., 2015*). Pleiotropic constraints limit evolution through complex and highly controlled global interactions of developmental processes, in which any disturbance would have great consequences (*Galis, Metz & Van Alphen, 2018*). Based on the same idea, *Riedl (1978)* proposed that some characters are strongly constrained (less evolvable), while others can change more freely (more evolvable). This difference relies on the burden of a character: as a structure evolves, it develops more relationships with other characters, becoming more and more interconnected and losing its freedom to evolve (*Riedl, 1978*). The burden theory states that the more the connections, the more the pleiotropic constraints. This can be easily interpreted from anatomical network parameters (*Rasskin-Gutman & Esteve-Altava, 2018*).

Among vertebrates, the anuran skeleton is especially interesting to be studied within an anatomical network approach because of their singular anatomy, topologically specialized for locomotion (*Dos Santos et al., 2017*). While the pelvic girdle and hindlimbs take on a significant role of propulsion, the pectoral girdle and forelimbs are mostly related to landing (*Emerson, 1979*; *Emerson, 1982*; *Nauwelaerts & Aerts, 2006*; *Astley & Roberts, 2014*). The skeletal pieces are affected by the high mechanical load of jumping locomotion, which could affect the genesis and development of sesamoids pieces (*Abdala, Vera & Ponssa, 2017*; *Abdala et al., 2019*). In this group, sesamoids are present mostly in the joints of the limbs and in the sacral vertebrae (*Hoyos, 2003*; *Ponssa, Goldberg & Abdala, 2010*;

PeerJ ______________________________________________________

*Abdala et al., 2019*). In particular, *Leptodactylus latinasus* (Leptodactylidae, Anura), our study case, has a total of 25 sesamoids (*Ponssa, Goldberg & Abdala, 2010*; *Abdala, Vera & Ponssa, 2017*), that were categorized as ESs and GSs following *Jerez, Mangione & Abdala (2010)*.

Based on the early and persistent intuition of many authors who worked on sesamoids, a driving hypothesis of our work was that sesamoids' nature relies profoundly on their connections. Thus, we propose to study them under an explicit network framework by modeling the *L. latinasus* skeleton. Based on this model, we explore two main questions related to the topological nature of sesamoids and their impact on the organization of the skeleton. Expressly: (1) How is the network structured? (2) Do the skeletal pieces categorized as embedded sesamoids, glide sesamoids and canonical skeletal elements differ in network parameters?

## MATERIALS & METHODS

### Sample and data acquisition

We examined six adult specimens of *L. latinasus* (one dissected and five cleared and stained). We complemented gross anatomy dissections with several previous anatomic descriptions (*Gaupp & Ecker, 1896*; *Dunlap, 1960*; *Nussbaum, 1982*; *Burton, 1998*; *Ponssa, Goldberg & Abdala, 2010*; *Abdala, Vera & Ponssa, 2017*). Due to the bilateral symmetry of the body, we built an adjacency matrix considering the right half of the body as a proxy of the whole configuration. An adjacency matrix defines the connectivity pattern of the anatomical network by indicating pairs of connected elements. Further details are available in Text S1 and Table S1.

### Network construction

Networks are appropriate mathematical models for tackling the study of biological systems because they are intrinsically collections of entities (nodes) connected through relationships (links). The identification of nodes and links is a critical issue that depends on what we want to know about the modeled system. We modeled the skeletal network of *L. latinasus* based on the main adjacency matrix. The network model was constructed to inquire on the topological nature of sesamoids within the skeletal system and the anatomical relation among them. Therefore, in our model the nodes represent canonical skeletal pieces and sesamoids (ossified, cartilaginous and fibrocartilaginous). The dorsal fascia was also modeled as a node due to its great extension and its role as insertion or origin point for muscles. Since joints, muscles, tendo-muscular units, tendons, and aponeurosis are functionally responsible for the flow of mechanical information among skeletal elements they were considered links in our study. Because all anuran sesamoids are postcranial, the consideration of the whole skull is not informative for our purposes, thus cranial bones were collapsed into a single node. We considered the network as undirected and weighted, considering weights as multiple links.

### Network modularity

A morphological module is a semi-independent set of densely interconnected skeletal pieces that are only sparsely connected to the rest of the skeleton (*Rasskin-Gutman &*

*Esteve-Altava, 2014*; *Dos Santos et al., 2017*). The number and composition of modules were identified using Order Statistics Local Optimization Method (OSLOM) (*Lancichinetti et al., 2011*). This community detection algorithm allows the detection of the statistical significance (*bs*) of modules with respect to random fluctuations. OSLOM takes into account the possibility of homeless nodes (i.e., nodes that are not assigned to any module), as well as the detection of overlapping modules, and the presence of hierarchical organization (*Lancichinetti et al., 2011*). For our analysis we followed the default options of the OSLOM algorithm, considering bs threshold as 0.1 and allowing the detection of singletons (homeless nodes). Because of the stochastic nature of the OSLOM modularity analysis, results may vary depending on the run (*Lancichinetti et al., 2011*; *Esteve-Altava, 2017*). We have reported the most stable modules among multiple runs.

### Sesamoid characterization
#### Node parameters
To investigate a possible difference among skeletal categories, we compared node centrality indicators taking into account three skeleton categories (CSE = canonical skeletal element, ES = embedded sesamoid, and GS = glide sesamoid). Centrality measures capture the relevance of the position of the individual nodes in the network (*Dos Santos et al., 2017*). The four most commonly used centrality measures were assessed: (1) Degree: the number of links of a node (*Csardi & Nepusz, 2006*); (2) Betweenness: the frequency of events in which a node is located in the shortest path between a pair of nodes (*Dos Santos et al., 2017*); (3) Closeness: the average length of the shortest path between that particular node and all other nodes in the network (*Freeman, 1979*); (4) Eigen-centrality: the first eigenvector of the adjacency matrix of the graph (*Bonacich, 1987*). Nodes with high eigenvector-centrality are those connected to many other nodes, which are, in turn, connected to many others (and so on). Central nodes, under this criterion, belong to centers of big cohesive sets of nodes (*Csardi & Nepusz, 2006*). It is important to notice that the centrality values of axial elements might be underestimated because our model included one half of the symmetric body. Lastly, we compared the averages of the aforementioned centrality parameters among the skeletal categories. We calculated the significance of the differences by Kruskal–Wallis tests and then we performed a post-hoc Wilcoxon pairwise comparison test. All the data and the code used to perform our analyses are available in Code S1 and Tables S3–S9. Code was written under R envirorment (*R Core Team, 2019*).

## RESULTS
### Network characterization
The anatomical network of *Leptodactylus latinasus* comprises 102 nodes connected by 328 physical connections (Figs. 1 and 2). Regarding centrality parameters, long bones (especially the tibiafibula), the ilium, and the dorsal fascia stand out by showing the highest centrality values (see Table S3). The module detection algorithm applied (OSLOM) revealed five significant and partial overlapping modules plus two singletons in the first hierarchical level (Figs. 1 and 2, Table 1): the **pectoral-forelimb module** (M1, $bs = 0.069$), the **axial-scapular module** (M2, $bs = 0.002$), the **axial-pelvic module** (M3, $bs = 0.081$),

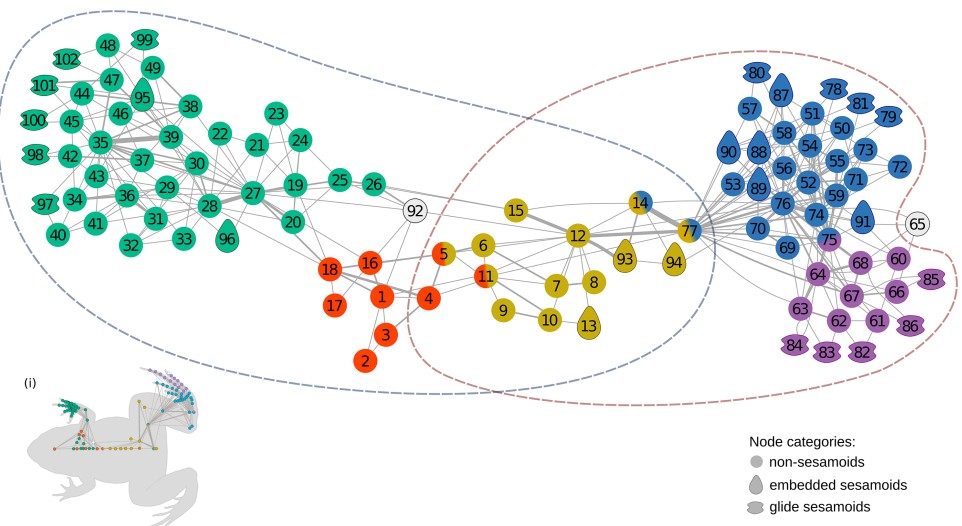

**Figure 1** **The anatomical network of *Leptodactylus latinasus* with an inset (i) providing a schematic representation of the network relative to the species body.** Links are weighted by the number of connections, and nodes are colored according to membership to modules: pectoral-forelimb module (green), axial-scapular module (orange), axial-pelvic module (yellow), hindlimb module (blue), IV–V toes module (purple), and homeless pieces (grey). Mix colored nodes are simultaneously members of two modules. Different shapes distinguish among skeletal categories of nodes: non-sesamoids, glide sesamoids, and embedded sesamoids. ID numbers are shown in Table 1.

the **hindlimb module** (M4, $bs = 0.063$), and the **IV–V toes module** (M5, $bs = 0.013$). The fourth vertebra and the urostyle (nodes IDs 5 adn 11) are simultaneously members of the two axial modules (M2 and M3). The femur and the ischium (nodes IDs 14 and 77) are simultaneously members of overlaps the axial-pelvic and the hindlimb module, and the fibulare (node ID 75) overlaps between the hindlimb (node ID 92) and the IV–V toes module. The fascia dorsalis and the third phalanx of digit V of the foot (node ID 65) were not assigned to any module in the first hierarchical level (singletons). The configuration of the second hierarchical level presented two modules with a broad overlap of the axial nodes. The third phalanx of digit V of the foot was also not included in any module in the second hierarchical level.

## Sesamoid patterns

Sesamoids are widely distributed through the network, being present in all modules except in the axial-pectoral (M2). GSs are arranged peripherally in the network topology. The pectoral-forelimb module (M1) includes two ESs and six GSs. In the axial-pelvic module (M3), there are three ESs. The hindlimb module (M4) shows the highest number of sesamoids (nine), and the IV–V toes module contains only GSs (five).

There are significant differences among node skeletal categories for the all centrality indicators measured. GSs exhibit the lowest average values for every centrality indicator, while CSEs exhibit the highest centrality indicator average values except for eigen-centrality. ESs do not differ significantly from CSEs in any of the centrality indicators. GSs are significantly lower both from SEs and from CSEs in degree and closeness. GSs do not differ

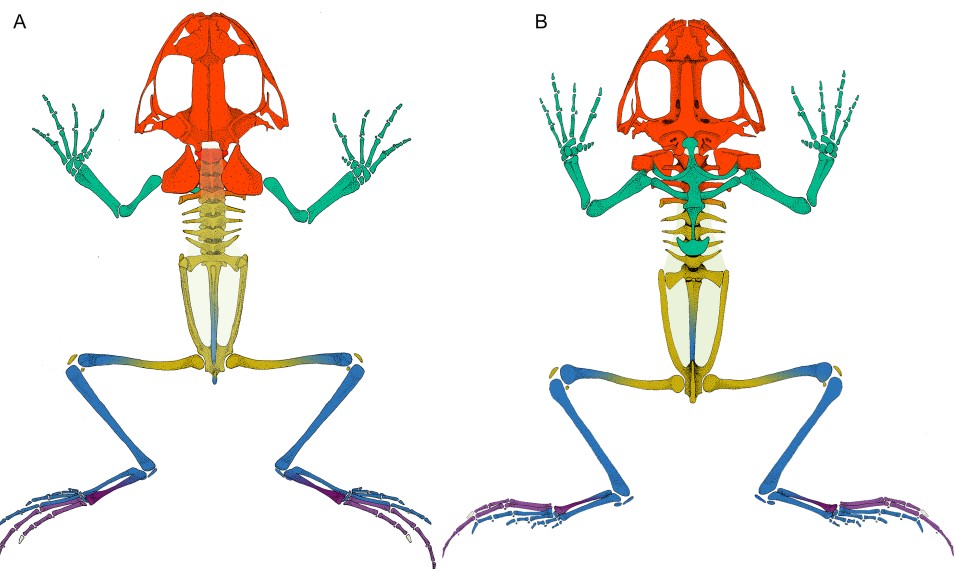

**Figure 2** ***Leptodactylus latinasus* skeleton with the pieces colored according to network modules.**
(A) Dorsal view. (B) Ventral view. The fascia dorsalis (pectoral-forelimb module) is represented in a non-saturated grey color to allow the visualization of overlapped structures. Color references: pectoral-forelimb module (green), axial-scapular module (orange), axial-pelvic module (yellow), hindlimb module (blue), IV–V toes module (purple), and homeless pieces (saturated grey). Mix colored pieces are simultaneously members of two modules.

significantly from ESs regarding betweenness centrality, but they do differ from CSEs in this parameter. GSs do not differ significantly from CSEs in eigen-centrality, but they do differ from ESs in this parameter (Fig. 3; Table 2; see also Table S2 and Fig. S1).

## DISCUSSION

Here we present a new approach to the study of sesamoids. Our results agree with the implicit assumptions of many previously proposed sesamoid definitions (see introduction). The connectivity patterns of the skeletal system reflect key properties of sesamoid identity, specially GSs, and suggest that they could have undergone a differential evolutionary mode.

### I. General network properties

In the first hierarchical level, and because the skull has been simplified to a single node, the modules are mainly related to the girdles and limbs. Anatomically, the pectoral girdle is divided in a ventral region, composed by the coracoid elements, and in a dorsal region, composed by the scapular elements (*Baleeva, 2009*). This regionalization could explain why the pectoral girdle is arranged in two modules: the axial-scapular module and the pectoral-forelimb module. In the axial-scapular module, the dorsal elements of the pectoral girdle are more connected to the cranium and to the first vertebrae than to the rest of the pectoral girdle. Indeed, both scapular elements and the cranium are connected by several muscles, such as the M. depressor maxillae which inserts in the lower jaw (*Ecker, 1889*; *Manzano, Moro & Abdala, 2003*). The cartilaginous connection between the scapular and

**Table 1  Modules composition.** ID node numbers are superscripted and elements highlighted in bold are included in more than one module.

| Module | Canonical skeletal elements | Embedded Sesamoid | Glide Sesamoid |
|---|---|---|---|
| Pectoral-forelimb (M1 - green) | clavicle[19], coracoid[20], episternum[21], omosternum[22], procoracoid[23], epicoracoid[24], mesosternum[25], xiphisternum[26], humerus[27], radioulna[28], radiale[29], ulnare[30], element Y $_{(f)}$[31], proximal prepolex[32], distal prepolex[33], carpal 2[34], carpal 3-4-5[35], metacarpal 2[36], metacarpal 3[37], metacarpal 4[38], metacarpal 5[39], FPI F2[40], FPII F2[41], FPI F3[42], FPII F3[43], FPI F4[44], FPII F4[45], FPIII F4[46], FPI F5[47], FPII F5[48], FPIII F5[49] | palmar sesamoid[95], pararadial[96] | metacarpal glide F2[97], metacarpal glide F3[98], metacarpal glide F4[99], interphalangeal glide FPII-I F4[100], metacarpal glide F5[101], interphalangeal glide FPII-I F5[102] |
| Axial-scapular (M2 - red) | cranium[1], atlas[2], V2[3], V3[4], **V4**[5], suprascapula[16], cleithrum[17], scapula[18], **urostyle**[11] | – | – |
| Axial-pelvic module (M3 - yellow) | **V4**[5], V5[6], V6[7], V7[8], V8[9], sacral vertebra[10], **urostyle**[11], ilium[12], **ischium**[14], pubis[15], **femur**[77] | sacral sesamoid[13], patella[93], graciella[94] | – |
| Hindlimb module (M4 - blue) | HPII F1[50], HPI F1[51], metatarsal 1[52], HPII F2[53], HPI F2[54], metatarsal 2[55], HPIII F3[56], HPII F3[57], HPI F3[58], metatarsal 3[59], distal prehallux[69], proximal prehalux[70], element Y$_{(h)}$[71], tarsal 1[72], tarsal 2-3[73], tibiale[74], **fibulare**[75], tibiofibula[76], **femur**[77], **ischium**[14] | cartilago sesamoide[87], plantar sesamoid I of the plantar aponeurosis[88], plantar sesamoid II of the plantar aponeurosis[89], OS Sesamoides tarsale[90], plantar sesamoid of the flexor digitorum[91] | metatarsal glide F1[78], metatarsal glide F2[79], interphalangeal glide HPII-I F3[80], metatarsal glide F3[81] |
| IV-V toes module (M5 - purple) | HPIV F4[60], HPIII F4[61], HPII F4[62], HPI F4[63], metatarsal 4[64], HPII F5[66], HPI F5[67], metatarsal 5[68], **fibulare**[75] | – | interphalangeal glide HPIII-II F4[82], interphalangeal glide HPII-I F4[83], metatarsal glide F4[84], interphalangeal glide HPII-I F5[85], metatarsal glide F5[86] |
| Singletons | fascia dorsalis[92], HPIII F5[65] | – | – |

**Notes.**

F, finger; FP, forelimb phalanx; HP, hindlimb phalanx; f, forelimb; h, hindlimb; T, toe; V, vertebra.

coracoid elements provides a considerable strength of connection while admitting some mobility and decreasing compression forces (*Baleeva, 2001*). The pectoral-forelimb module is formed by the coracoid elements and the forelimb. This region has an important and complex function absorbing the stress of the impact in the landing phase of the jump (*Emerson, 1982*). Also, both modules are involved in the complex mechanism of anuran acoustic perception (*Lombard & Straughan, 1974*; *Kardong, 2012*).

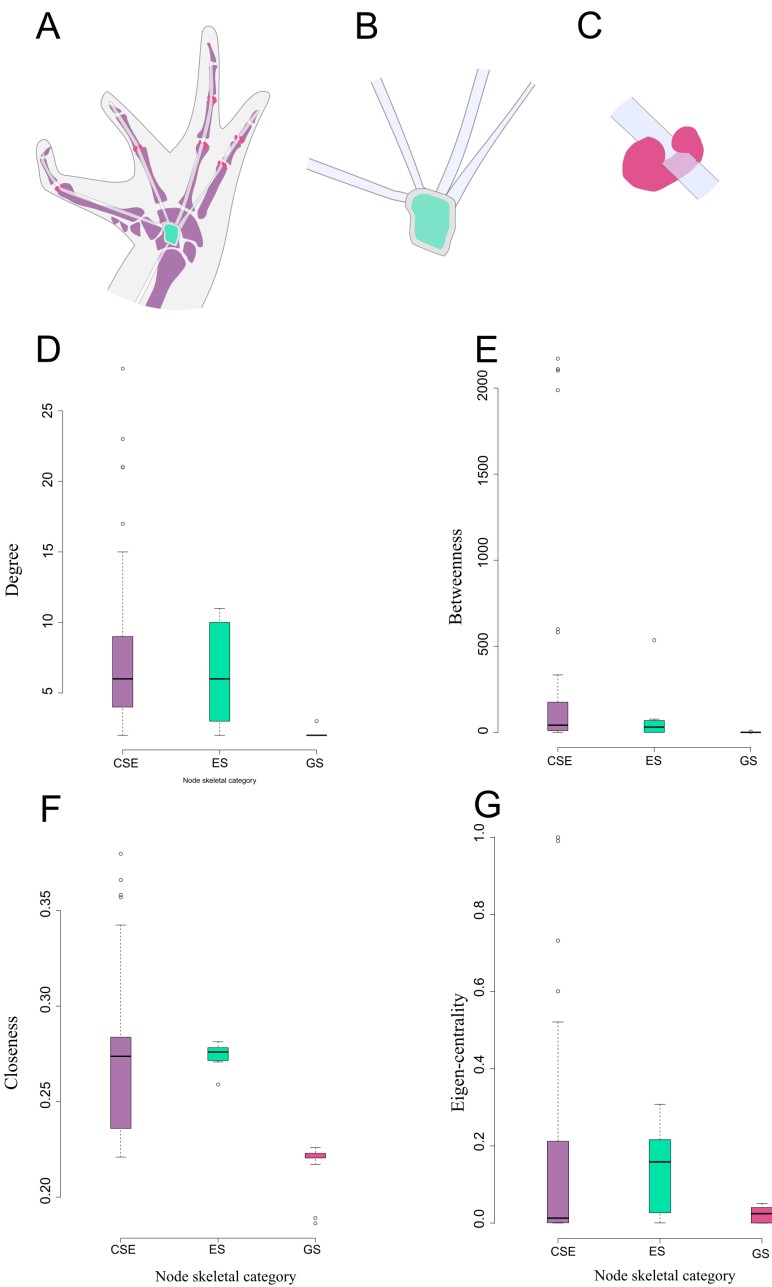

**Figure 3  Schematic representation and node parameter values of each node type in the network.** (A). Ventral view of the left hand of *Leptodactylus latinasus* showing the canonical bones (light purple), the palmar sesamoid (light blue) interphalangeal and metacarpophalangeal glide sesamoids (pink). (B). Detail of the palmar sesamoid (embedded type) with the flexor tendons of the digits. (C). Detail of an interphalangeal glide associated with a flexor tendon. (D). Boxplots of degree values by node category. (E). Boxplots of betweenness values by node category. (F). Boxplots of closeness values by node category. (G). Boxplots of eigen-centrality values by node category. Abbreviations: CSE: Canonical Skeletal Element (light purple); ES: embedded sesamoid (light blue); GS: glide sesamoid (pink).

The axial-pelvic module is where most of the overlapping occurs in both hierarchical levels of our modularity analysis. This particular configuration of the anuran pelvic girdle seems to represent a topological transitional region between an anterior and posterior region of the network. The elements of this module are deeply affected by developmental changes during metamorphosis from a swimming tadpole into a tailless jumping adult (*Soliz & Ponssa, 2016*). These transformations result in a compact and highly integrated vertebral column plus pelvic girdle, epitomized in the network by the membership of the 4th vertebra and the urostyle to both the axial-pectoral and axial-pelvic modules. The femur and the ischium are also simultaneously members of two modules, the axial-pelvic and the hindlimb module. Previous studies have shown that, at the beginning of embryonic formation, the femur develops in close contact with the future acetabulum of the pelvic girdle (*Pomikal, Blumer & Streicher, 2011*). Furthermore, functional co-dependencies exist in the pelvic-hindlimb boundary, in which sequential movement of the hip and leg joints were identified during a typical frog jump (*Astley & Roberts, 2014*; *Nauwelaerts, Stamhuis & Aerts, 2005*).

The hindlimb module is constituted by pieces of the stylopodium, zeugopodium, and autopodium, most of them are long bones with high centrality values. Long bones have been recurrently recovered as central nodes in network analysis of tetrapod limbs (*Diogo et al., 2015*; *Dos Santos et al., 2017*). The zeugopodization hypothesis in anurans postulates the elongation of the tibiale and fibulare and a consequent distal shift in the zeugo-autopodial border (*Diogo & Ziermann, 2014*; *Dos Santos et al., 2017*). This could explain why the tibiale and fibulare show high centrality values, that in turn, could be associated with their functional importance as an extra site for muscle attachment (*Handrigan & Wassersug, 2007*). Moreover, the fibulare is also part of the IV–V toes module, which is composed of those elements of the foot aligned with this bone. The elements of IV–V toes module also share a common origin of their flexor tendons, which arise from the flexor digitorum brevis superficialis. Contrary to those of the toes I–III, which originate from the aponeurosis plantaris (*Gaupp & Ecker, 1896*).

## II. Sesamoid Nature, as revealed by node parameters

Sesamoids were often conceptually placed outside or beyond the skeleton (*Vickaryous & Olson, 2007*). *Diogo et al. (2015)*, indeed modeled many sesamoids as isolated nodes (i.e., without any connection to other pieces) in their strictly skeletal network. On the contrary, our anatomical network does include sesamoids as inner pieces of the skeleton, linked by tendons and muscles to the other skeletal elements. Surprisingly, all of them are unambiguously integrated within four of the modules of the first hierarchical level of the network, being absent only in the axial-pectoral module. Connectivity patterns arrange GSs as peripheral elements of the system, while the ESs are more variably distributed through the network. Both are, in general terms, weakly connected with the canonical pieces of the skeleton as recovered by the centrality indicators. In this sense, sesamoids seem to be lowly burdened structures, *sensu Riedl (1978)*. Connections between elements are established during embryological development (*Rasskin-Gutman & Esteve-Altava, 2018*). Sesamoids develop independently and relatively late in comparison to other skeletal

Fontanarrosa et al. (2020), *PeerJ*, DOI 10.7717/peerj.9691

**Table 2 Sesamoid centrality indicators summary.**

| Sesamoids | ID | Muscle/tendon | Cat. | D | C | B | E | M |
|---|---|---|---|---|---|---|---|---|
| Cartilago sesamoide | 87 | plantaris profundus | Embedded | 6 | 0.272 | 4.183 | 0.157 | 4 |
| Glide interphalangeal II-I of finger IV | 100 | interphalangeal | Glide | 2 | 0.189 | 0.000 | 0.000 | 1 |
| Glide interphalangeal II-I of finger V | 102 | interphalangeal | Glide | 2 | 0.186 | 0.000 | 0.000 | 1 |
| Glide interphalangeal II-I of toe IV | 83 | flexor digitii brevis | Glide | 2 | 0.221 | 0.467 | 0.031 | 5 |
| Glide interphalangeal II-I of toe V | 85 | flexor digitii brevis | Glide | 2 | 0.221 | 0.583 | 0.024 | 5 |
| Glide interphalangeal II-Iof toe III | 80 | flexor digitii brevis | Glide | 2 | 0.221 | 0.726 | 0.025 | 4 |
| Glide interphalangeal III-II of toe IV | 82 | flexor digitii brevis | Glide | 2 | 0.217 | 0.000 | 0.013 | 5 |
| Glide of metacarpal II | 97 | lumbricalis brevis | Glide | 2 | 0.221 | 0.000 | 0.000 | 1 |
| Glide of metacarpal III | 98 | lumbricalis brevis | Glide | 2 | 0.221 | 2.042 | 0.000 | 1 |
| Glide of metacarpal IV | 99 | lumbricalis brevis | Glide | 2 | 0.221 | 1.930 | 0.000 | 1 |
| Glide of metacarpal V | 101 | lumbricalis brevis | Glide | 3 | 0.226 | 4.404 | 0.000 | 1 |
| Glide of metatarsal I | 78 | flexor digitii brevis | Glide | 2 | 0.223 | 0.000 | 0.051 | 4 |
| Glide of metatarsal II | 79 | flexor digitii brevis | Glide | 2 | 0.223 | 0.309 | 0.047 | 4 |
| Glide of metatarsal III | 81 | flexor digitii brevis | Glide | 2 | 0.224 | 0.610 | 0.044 | 4 |
| Glide of metatarsal IV | 84 | flexor digitii brevis | Glide | 2 | 0.225 | 0.450 | 0.045 | 5 |
| Glide of metatarsal V | 86 | flexor digitii brevis | Glide | 2 | 0.221 | 1.444 | 0.037 | 5 |
| Graciella | 94 | gracilis major | Embedded | 3 | 0.271 | 76.943 | 0.097 | 3 |
| OS sesamoides tarsale | 90 | Achilles tendon | Embedded | 7 | 0.277 | 4.804 | 0.216 | 4 |
| Palmar sesamoid | 95 | Flexor digitorum communis - Flexor plate | Embedded | 10 | 0.275 | 535.513 | 0.001 | 1 |
| Pararadial | 96 | Extensor carpi radialis | Embedded | 2 | 0.259 | 0.000 | 0.001 | 1 |
| Patella | 93 | knee aponeurosis | Embedded | 3 | 0.278 | 0.000 | 0.160 | 3 |
| Plantar sesamoid of the flexor digitorum | 91 | flexor dig brevis superficialis | Embedded | 6 | 0.273 | 68.815 | 0.167 | 4 |
| Plantar sesamoid I of the plantar aponeurosis | 88 | plantar aponeurosis | Embedded | 11 | 0.281 | 61.837 | 0.297 | 4 |
| Plantar sesamoid II of the plantar aponeurosis | 89 | plantar aponeurosis | Embedded | 11 | 0.281 | 56.528 | 0.308 | 4 |
| Sacral sesamoid | 13 | internal ligament of the sacrum | Embedded | 2 | 0.277 | 0.000 | 0.027 | 3 |

**Notes.**

Cat, Sesamoid category following *Jerez, Mangione & Abdala (2010)*; **M**, Module; **D**, Degree; **B**, Betweenness; **C**, Closeness; **E-C**, Eigen centrality.
elements, only later becoming associated with the primary skeleton (*Hall, 2005*; *Vera, Ponssa & Abdala, 2015*). Although sesamoids may have an imperfect fossil record, probably due to their loose connectivity pattern (this work), their earliest fossil reports predate the Jurassic period (200+ mya) (*Vickaryous & Olson, 2007*). By then, most of the skeletal pieces would have been evolving for at least 420 million years (*Ravi & Venkatesh, 2008*). In turn, sesamoids' high diversity in size, shape, number, and distribution (*Abdala et al., 2019*) could be a consequence of being low burdened structures. In fact, high rates of evolution of sesamoid bones were reported by *Baum & Smith (2013)*. Thus, the loose connectivity pattern characterizing sesamoids seems to be a consequence of their delay in ontogeny and phylogeny.

The facultative expression of many sesamoids in the phenotype as a response to continuous mechanical stress (i.e., epigenetic influence) (*Abdala & Ponssa, 2012*; *Abdala et al., 2019*) could be, at least in part, a consequence of the low burden of these skeletal pieces. Perturbations on sesamoids development is unlikely to be accompanied by systemic consequences for an organism, as could be the case of perturbations on the development of canonical elements. As lowly constrained pieces, sesamoids may be labile evolutionary elements, with a relatively high capacity of generating heritable phenotypic variation (*Kirschner & Gerhart, 1998*), in turn the path of the appearance of evolutionary novelties would be facilitated. This rationale is in accordance with the dynamic model stated by *Abdala et al. (2019)* in which sesamoids are proposed as a source of new skeletal morphologies available to natural selection processes.

## II-1. Embedded sesamoids

ESs centralities values were not significantly different from CSEs elements, but they resulted to be significantly more central than those of the GSs (except for betweenness). The fact of being *embedded* in the connective tissue of the most powerful muscles of the limbs (*Jerez, Mangione & Abdala, 2010*), which were considered as network links, straightforwardly contributes to the higher centrality values of ESs when compared to GSs. ESs are distributed in three modules related to the limbs and the pelvic girdle (M1, M3, and M4), and absent from axial-scapular and IV–V toes modules (M2 and M5). Most ESs are included within the hindlimb module, coincidentally, this module is subject to the highest mechanical forces during the take-off phase of the jump (*Nauwelaerts & Aerts, 2006*). The palmar sesamoid (pectoral-forelimb module) showed a notably high betweenness value among embedded sesamoids, and surprisingly similar to top-ranked canonical elements (Table S3). This could be associated with the fact that the palmar sesamoid is embedded in the m. flexor digitorum longus which is the source of the flexor tendons of digits II–V (*Ponssa, Goldberg & Abdala, 2010*; *Diogo & Abdala, 2010*).

It is logical to think that nearby pieces will tend to be more connected than distant pieces. Thus, a correspondence between network modules and euclidean regions of the body is expected (*Dos Santos et al., 2017*). Sesamoids, in general, have links other than joints connecting them to other skeletal pieces (*Vickaryous & Olson, 2007*; Table S1). This property allows them to defy the general proximity imposition, in such a way that they are able to share a module with spatially distant pieces. In fact, the patella and the

graciella sesamoids, located in the knee joint, are co-opted by a more proximal module (axial-pelvic module) instead of the hindlimb module, as we could expect following a spatial neighborhood criterion. These sesamoids constitute the only elements in the network with such kind of behavior. This pattern could be explained by their remote connection with the pelvic girdle by the cruralis and the gracilis major muscles, respectively, which form a set of muscles required for the extension and flexion of the knee joint (*Abdala, Vera & Ponssa, 2017*; Table S1). Additionally, *Eyal et al. (2015)* show that, in mice, the patella arises as part of the femur but from a distinct pool of progenitors. Thus, probably, the patella membership to the axial-pelvic module can be explained by complex cellular and genetic mechanisms during the morphogenesis process.

## II-2. Glide sesamoids

Centrality indicators mainly segregated the GSs from the other skeletal categories. Frequently, GSs are implicitly excluded from sesamoid conceptual delimitation, due to def-initions typically consider sesamoids as elements surrounded by tendinous or ligamentous structures (e.g., *Hall, 2005*); "(...) *sesamoids are independent ossifications/chondrifications within tendons*"). Moreover, developmental evidence has shown that although ESs and GSs share the same progenitor cells, they have different developmental signaling paths (*Eyal et al., 2019*).

Glide sesamoids are significatively less connected than ESs and CSEs when comparing degree and closeness. A different trend is revealed by the eigen-centrality indicator, which is similar between ESs and CSEs, but distinguishes the two sesamoid categories highlighting the particularities of GSs. Low eigen-centrality indicates that not only GSs, but also that their neighbor nodes have few connections. The unusually low centrality indicators of GSs could be a proxy of a high evolvability of those bones following the burden theory (*Rasskin-Gutman & Esteve-Altava, 2018*). Indeed, high intraspecific variation in number and morphology has been reported in glides (*Ponssa, Goldberg & Abdala, 2010*). Therefore, low connectivity could represent an alternative strategy to modularity in order to increase evolvability.

The anatomical distribution, shape, constitution, and the paired condition of GSs of the forelimb in *L. latinasus* (*Ponssa, Goldberg & Abdala, 2010*), is similar to those of paraphalangeal elements that characterize many pad-bearing geckos (Squamata). The multiple origins of paraphalanges plus their extremely variable morphology (*Wellborn, 1933*; *Russell & Bauer, 1988*; *Gamble et al., 2012*; *Fontanarrosa, Daza & Abdala, 2018*) supports the idea of their lability in evolutionary terms. Curiously, lizards that lack paraphalanges also lack GSs related to interphalangeal joints (*Fontanarrosa, 2018*). Additional network analysis, modeling a species with paraphalanges, would most likely indicate that they are relatively disconnected structures of the main skeleton. Connectivity patterns have long been a criterion for the recognition of homologies (*Geoffroy Saint-Hilaire, 1818*). Thus, identifying common connections to specific elements, could reveal putative homologous structures through distant related lineages such as paraphalanges and GSs. Furthermore, dissimilar connectivity patterns between ESs and GSs found in this study suggest that they may not be members of the same hierarchical category. Future

studies based on complementary sources of evidence, such as development or evolution, are required to test this hypothesis.

## CONCLUSIONS

Here we presented a new approach to the study of sesamoid identity and we hope to contribute to the current research on their morphological evolution. Our findings raise interesting questions to be investigated in other species of tetrapods, as well as by complementary areas of research, such as developmental or evolutionary biology. Multiple sesamoid definitions based on their relations with canonical bones and connective tissue were calling for their explicit framing under network theory. After performing an anatomical network analysis of a model anuran species (*L. latinasus*), we inquired first on the general topology, and more specifically on sesamoid connectivity patterns. The main conclusions that emerged from this approach are:

1. The skeletal elements were clustered in five modules that reflect a functional organization. Four modules contain at least one sesamoid.

2. Sesamoids, in general terms, are peripheral elements of the network, with few connections to the canonical skeleton. This could explain their considerable variation on size, shape, number, distribution and high evolvability. These results support the hypothesis of sesamoids as morphological innovations generators.

3. Embedded sesamoids have, on average, similar centrality values to the canonical skeletal elements. These sesamoids are surrounded by connective tissue, thus are prone to have more connections than glide sesamoids. While glides are adjacent to tendons, but not fixed to them.

4. Glide sesamoids have the lowest values for every centrality indicator measured, when compared to the other skeletal categories.

5. Similarities between embedded sesamoids and canonical bones, in addition to glides' own singularities, leave an open question as to whether all embedded and glide sesamoids have the same nature.

## ACKNOWLEDGEMENTS

Conversations and exchanges of ideas with Virginia Abdala (IBN, CONICET, Argentina) helped us considerably. We also thank Borja Esteve-Altava, Julio Mario Hoyos, and an anonymous reviewer for their suggestions to improve the manuscript. We thank Diego Baldo, curator of the herpetological collection of the Laboratorio de Genética Evolutiva of Instituto de Biología Subtropical, for providing the study material.

### Funding

Gabriela Fontanarrosa, Jessica Fratani and Miriam C. Vera are supported by postdoctoral fellowships (CONICET) and PIP0389, PICT 2016-2772, and PICT 2018-0382 funds. The

funders had no role in study design, data collection and analysis, decision to publish, or preparation of the manuscript.

## Grant Disclosures
The following grant information was disclosed by the authors:
CONICET.
PIP0389, PICT 2016-2772, and PICT 2018-0382.

## Competing Interests
The authors declare there are no competing interests.

## Author Contributions
- Gabriela Fontanarrosa and Jessica Fratani conceived and designed the experiments, performed the experiments, analyzed the data, prepared figures and/or tables, authored or reviewed drafts of the paper, and approved the final draft.
- Miriam C. Vera conceived and designed the experiments, performed the experiments, prepared figures and/or tables, authored or reviewed drafts of the paper, and approved the final draft.

## Data Availability
Specimens examined first-hand are listed by accession numbers. Dissected specimen: MCV 364. Cleared and stained specimens: LGE 12248, LGE 12128, LGE 12139, LGE 12121 and LGE 12129. Institutional Abbreviation: LGE, Laboratorio de Genética Evolutiva; MCV, Mirian C. Vera's field number. Specimens were observed under a stereomicroscope Zeiss.

The raw data and code are available in the Supplementary Files.

## Supplemental Information
Supplemental information for this article can be found online at http://dx.doi.org/10.7717/peerj.9691#supplemental-information.

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
