# Peer review of "Delimiting the boundaries of sesamoid identities under the network theory framework"

_PeerJ, doi:10.7717/peerj.9691_

## Round 0.1 · original submission · Major Revisions

Three reviewers have given expert reviews, and all of them are constructive but also collectively they raise a large number of points that would require extensive revisions to address. These points range from more minor presentational ones to substantial, technical issues of methods and concepts. We will need re-review of this MS by at least 2 of the reviewers. Please do take your time in revising; we know it is a difficult period for many people in the world. Any "deadlines" are purely suggestions.

·

Basic reporting

The authors present an interesting comparison of the many sesamoid bones in an anuran skeleton. They considered the topological place of sesamoids within the context of the entire skeleton, finding that these skeletal or cartilaginous elements show some connectivity features that set them apart from other skeletal elements.

The introduction and discussion provide adequate context for the study. If anything, I would suggest citing the work of Étienne Geoffroy Saint-Hilaire, who proposed the principle of connections as a rule to identify same structures in different organisms (i.e. homology in modern language) instead of Gutman & Esteve-Altava (2014) in Line 349.

I would suggest revising the writing style to make some ideas clearer. At times, there are unnecessarily complicated sentences. For example, [line 317] “Joints impose spatial neighborhood relationships between the skeletal pieces”, where I guess the meaning is ‘Joints connect nearby skeletal pieces’. Or [line 319] “Sesamoids as skeletal pieces that are, in general terms, characterized by not being articulated to other pieces of the skeleton,” which either is missing some verb (?) or would mean ‘Sesamoids are skeletal pieces that do not articulate to other skeletal pieces’.

Experimental design

The driving hypothesis is interesting and justifies using a network-based analysis. However, I have some concerns about the specific questions: 1b [line 87] “Is it possible to identify anatomical modules in the network?” and the cross-testing of 2a [89] and 2b [91].

Regarding question 1b. The community detection algorithm used will find modules even when a modular partition makes no sense or is not ‘truly’ present, such as in a random network or in a regular network. For example, if one uses this algorithm on a regular ring network of 10 nodes, cluster_optimal(make_ring(10)), it produces an output with 3 modules and a fairly good modularity quality value (Q=0.36). Because of that, we know a priori that the answer to this question is yes.

There are other community detection methods that will provide a better answer to this question, for example, OSLOM (http://www.oslom.org/index.html). This method has the potential to unveil also non-modular networks, as well as other features that would be interesting evaluating here, for example, if sesamoids contribute to the overall modularity of the skeletal network or are singletons without contribution. References to this algorithm are available it the software webpage. An example of its use and interpretation in anatomical networks is available in https://doi.org/10.1002/jmor.20690.

If authors decide to use algorithms based on maximizing modularity Q value (like cluster_optimal), then it would be necessary to estimate the error of Q to know if the parition is of enough quality. This can be done using a jackknife procedure where each connection is treated as an independent observation. Details about estimating Q error are available in https://doi.org/10.1002/jmor.20690 or https://doi.org/10.1103/PhysRevE.69.026113

Regarding questions 2a and 2b. The authors test whether non-sesamoid and sesamoid bones (on the one hand) and non-sesamoid vs embedded sesamoids vs glided sesamoid (on the other hand) show different values for network parameters. I have different concerns here with the statistical testing. The first one is that the testing of eigen-centrality is omitted (Supplemental Table S2), but I suppose that they also tested it. Why are eigen-centrality tests not shown? The second one is about the type of tests selected (Kruskal-Wallis tests and Wilcoxon). The authors do not justify the reason to mix both types of tests. Why not a partially nested ANOVA? Moreover, because multiple tests are performed (on various overlapping groups and for 3-4 variables each), I believe a correction for multiple testing is necessary (e.g. Bonferroni or similar). I suggest that this part of the study is evaluated by a professional statistician.

Regarding the building of the anatomical network. It is not clear why the authors decided to [line 98] “built an adjacency matrix considering the whole skeleton of the right half of the body” only. Because they are including many skeletal elements aligned in the midline of the body plan (e.g. crania, vertebra), making only a network of half the body has a big impact on the output of the analysis. For example, midline elements will be half important (half central) if we only analyze half the body. This can also impact modularity results: as the left-right axis is omitted, modularity will emphasize the A-P axis. What is the reason to make a network of half the skeleton and not of the whole skeleton?

Finally, I found a problem in the R script for the analysis that may have had a substantial impact in the measurement of some network variables. The authors load their raw edge list data from an .csv containing all the connections between elements. Some of these pairs of elements have more than 1 connection of a different type (e.g. V3 and V4 have an articulation contact and a muscular contact). Then, they used the function graph.edgelist to create the main network for analysis. However, this network is not simple and contains multiple links (this can be checked with function is_simple). Multiple links have an impact on the quantification of parameters density, degree, betweenness centrality, and eigen centrality. For example, edge_density function (to calculate density) compares actual number of connections with the maximum theoretically possible; however, if multiple links are allowed, then the maximum becomes infinite. Consequently, density is overestimated. Some of the functions used to calculate other parameters may also have similar problems.

The details in the Methods section suggest that the authors are not aware of this issue. For example, in line 110, they state that “A weighted version of the network serves as a simpler graphical display in figure 1”. I guess this network weights come from the multiple links (2 links = weight value of 2).

To solve this problem, I suggest either working with the simplified network without multi-links or using the weighted version of the network explicitly. Both actions will change the measurements for some of the parameters. There is a third way. Authors could acknowledge that they are using a multigraph and analyze it accordingly with specific algorithms for this type of networks.

Validity of the findings

Because of the methodological problems mentioned, I am not certain whether the findings of this study will hold.

Additional comments

This study addresses very interesting questions: (1) what is the topological nature of sesamoids? And (2) what is their impact on the organization of the skeleton? An anatomical network analysis is the right tool to answer these questions. However, to answer these questions properly, the authors need to address important methodological issues regarding the building of the network and the algorithms used to analyzed it.

Reviewer 2 ·

Basic reporting

1. More context is needed in the intro. Why are anurans a good system to study sesamoids (73-81)? “A driving hypothesis of our work was that sesamoids' nature relies profoundly on their connections” (lines 83-84) – what evidence supports this hypothesis?
2. In discussion, compare with results of other network analyses that included sesamoids, e.g. Diogo et al. 2015
3. 371-372: “Beyond their intrinsic morphological properties, connectivity patterns incorporate key aspects of sesamoid identity.” This statement (and others elsewhere in the ms) implies that connectivity could be useful in identifying and/or defining sesamoids, but this idea is never explicitly stated.
4. 357-358: Are the authors arguing that glide sesamoids are not sesamoids? Please clarify.
5. The article is clearly structured and the figures and underlying data make sense. However, the language requires editing throughout to correct grammatical mistakes and improve clarity. In particular, parts of the introduction and discussion are very difficult to follow (e.g., 317-325).

Experimental design

1. The investigation seems to be technically valid and well done, and the methods are described sufficiently. However, I have a serious concern about the choice of methodology, which treats muscles as links (lines 106-107). As far as I’m aware, the existing literature treats muscles as nodes, not links (e.g. Diogo et al. 2018, Molnar et al. 2017 – cited in the current ms) or does not consider them at all (e.g., Esteve-Altava et al. 2013). This may be a perfectly valid choice, but there is no explanation or justification offered. Furthermore, this difference must be taken into account when comparing the results with previous studies that used different methods (e.g., lines 241, 245).
2. 86-87: Research questions 1a and 1b do not really add anything to the study

Validity of the findings

1. The conclusions are well supported by the data. I do have several suggestions about how the conclusions are stated: first, I think there are too many conclusions and that it would be better to number them than use bullet points. I would remove points 1 and 3 and combine points 5 and 6. Also make sure they are clearly linked to the original research question.
2. Line 368: “antero-posterior differentiation along the vertebral axis” is a confusing phrase. Do you mean differences in network structure between the pectoral and pelvic limbs?

Additional comments

The manuscript describes an anatomical network model of the (postcranial) musculoskeletal system of the frog Leptodactylus latinasus and its implications for the evolution of sesamoids. L. latinasus has an unusually large number of sesamoids in the hindlimb compared to other tetrapods. The study found that sesamoids, particularly gliding sesamoids, have low connectivity, possibly making them more evolutionarily labile. Exploring sesamoids from a network perspective is an interesting idea, but I have concerns about the methodology and I feel that more context and clarity are needed in the introduction and discussion.

·

Basic reporting

• Clear and unambiguous, professional English used throughout.

Although I am not completely competent to criticize the use of English, from my experience it seems to me to be fine, without apparent deep grammatical errors that make reading unintelligible.

• Literature references, sufficient field background/context provided.

Yes, very good literatue with a contest and suffiicent background.

• Professional article structure, figures, tables.

It seems to me a well written paper. The parts it is made of, are clearly structured, easy to follow. In general, then, it seems to me a pleasant and very interesting article, with great possibilities of being imrpoved, both in its results, and in both its discussion and conclusions (see below my observations on this matter). Figures and tables are clear.

Experimental design

• Original primary research within Aims and Scope of the journal.

Yes.
• Research question well defined, relevant & meaningful. It is stated how research fills an identified knowledge gap.

Yes. The questions they pose are relevants. In fact, it is interesting that they pose many questions in an explicit way.

These are the questions they posed:

Can embedded and glide sesamoids, be defined under the same criteria?:
I did not see the answer to this question in the text

1.a) How is the network structured?:

Answered in the lines 160 y 166.

1.b) Is it possible to identify anatomical modules in the network?:

Answered in the lines 169 y 178.


1.c) Is the antero-posterior differentiation of the anuran skeleton reflected in their element connectivity pattern?:
Answered in the Modularity and sub.networks comparisons items

and the last ones relating to sesamoid interactions within the network: Answered in the sesamoid patterns item

2.a) Do sesamoids share common network properties that make them distinguishable from other skeletal pieces?: Answered in the sesamoid patterns item

2.b) Do embedded and glide sesamoids differ in network parameters? Answered in the sesamoid patterns item, and in the last conclusion.


• Rigorous investigation performed to a high technical & ethical standard.

Yes

• Methods described with sufficient detail & information to replicate.

Although there is not, strictly speaking, an experimental design, there is a methodology of how they did to develop the modules and anatomical networks. However, I still have the question of whether they only used the R package to build both the modules and the anatomical networks, or they also used, for example, a main components analysis (PCA), in addition to other statistical packages that were not explicit in the methodology. In summary, I think the methods for these anatomical network and module constructions should be more detailed (see, for example: Evolutionary parallelisms of pectoral and pelvic network-anatomy from fins to limbs. Science Advances 08 May 2019: Vol. 5, no. 5, eaau7459 DOI: 10.1126/sciadv.aau7459).

Validity of the findings

• Impact and novelty not assessed.


I think there is not enough evaluation of the impacts and news generated by this study. For example, in developmental biology and systematics. Although they include something, they could be expanded This article shows that there are cases when a clear independence of the characters cannot be shown. The study shows that there is interdependence between sesamoid elements in the construction of modules and anatomical networks. We see it when they say "sesamoids nature relies deeply on their connections". Where is the proof of this hypothesis? Where's your answer? What would be its systematic application, even if it is from a functional point of view?
Although it is based on a species, could these modules and networks be extrapolated to other species of anurans, or can it only be per species?
It would be possible to make comparisons with studies in mammals and reptiles, for example, and even humans?

• All underlying data have been provided
Yes. Maybe a short description of the sesamoids is missing, taking into account that they affirm the importance of their morphology for the construction of networks and modules.
Although in Ponssa et al.(2010) there is a description of sesamoids in this species, I think it would be important to highlight here, in a synthetic way, the fundamental elements of this description that are the basis for the construction of modules and networks. I say this taking into account that, in line 34, they stated that “The loose connectivity pattern of sesamoids could be related to their evolvability, which in turn seems to be reflected in their morphological diversity and their facultative expression”. How does morphological diversity explicitly influence these constructions?

• All underlying data are robust, statistically sound, & controlled.

Apparently yes, but they need to deep in the methodology about both the modules and antomical networks construction.
• Conclusions are well stated, linked to original research question & limited to supporting results.

Yes, they are well stated and linked to the research. However,
I cannot see where this conclusion comes: “These results support the role of sesamoids as morphological innovations generators”.

I think that they need to respond in the conclusions to the questions and hypotheses they did in the introduction. Although the answers are included in t he discussion, I think they should be made explicit in the conclusions.

Additional comments

The study of the sesamoid elements is still very important, both from a morphological and functional point of view. However, little has been explored compared to other parts of the skeleton of tetrapod vertebrates, such as the skull, spine, or limbs. As they are structures that are mostly considered floating, sometimes difficult to observe, and apparently distributed without patterns, they give the impression that they are of little use in solving biological questions. I think that studies like this, in which it puts a different context for its application in biology, should be used much more to try to solve problems of kinship relationships, of the evolution of frogs, of their biomechanics, of the biology of development, etc. I hope the authors maintain this line of research.

---

## Round 0.2 · Minor Revisions

Well done- the reviewers are pleased with revisions and there are just some wording/structural changes suggested that should help improve the study. We look forward to the revised MS.

·

Basic reporting

The authors have made a thoughtful revision of the analyses in first manuscript following all my suggestions. I am fully satisfied with the results and I have no further comments.

Experimental design

In my opinion the new, revised analyses are sound and make the overall results stronger. I do not have further comments.

Validity of the findings

The now revised results are more compelling. The fact that they do not change substantially after improving the methods mean that the authors were already into something with a real biological meaning. Now, better supported in the revised version. I have no further comments.

Additional comments

I congratulate the authors on a very thoughtful and through revision of the methods and analyses used. I rarely see authors acknowledging and amending these problems when they are made aware of them. Also, the manuscript now is much easy to read and figures look nicer (specially Figure 1).

Reviewer 2 ·

Basic reporting

My concerns about the introduction have been addressed, and it is clear and informative. The language has been improved throughout, and additional relevant comparisons with previous studies have been added. I have a few minor questions/suggestions: 1-4 are about the interpretation of results, while 5-10 are just about wording.

1. Line 239: please make it even clearer here how your methods differ (e.g., which studies did not model muscles)
2. Lines 81-82 “sesamoids reflect an antero-posterior differentiation pattern, being present mostly in the joints of the limbs and in the sacral vertebrae” – please clarify how presence in the limb joints and sacral vertebrae relates to the anterior-posterior axis.
3. 193-194: “The network modules reflect a functional clustering pattern, instead of a topological organization constrained to element's proximity.” Please justify this statement. It looks like most elements are grouped with other nearby elements, and it is stated later in Results that the patella and graciela sesamoids are the only elements in the network that do not follow a spatial neighborhood criterion.
4. 324-325 “connectivity similarities among paraphalanges and GSs could indicate that they may share a deep common origin.” I think this requires further justification. As I understand it, Saint-Hilaire was talking about connections to specific elements, while in this instance you simply have two types of elements that are loosely connected to the main skeleton.
5. Abstract: “physical connections were considered links” – this is ambiguous. How about “muscles and bone to bone articulations were considered links” or similar?
6. Lines 110-112 “and tendinous fascia (in case of not containing a sesamoid embedded)” Unclear what is meant here. Please rephrase.
7. Lines 112-113 “The links are joints, muscles, tendo-muscular units, tendons, and aponeurosis (in case of containing a sesamoid embedded).” This is a bit confusing. An example might help. (this would also help to explain why you chose to model muscles as links!)
8. Line 156: especially (not specially)
9. 265-266 “The above exposed is in tune” rephrase
10. Line 316 “reminds those” Rephrase

Experimental design

The research questions have been condensed and clarified, and the authors have expanded on the reasoning behind their methodological choices. However, I still would like to see a fuller discussion of the latter issue. First, you might say explicitly in the manuscript what was stated in the rebuttal letter: that previous studies coded muscles differently, and why you chose to do it this way. Second, for the purposes of studying sesamoids, why is your approach better than coding muscles as nodes? You need muscles to study sesamoid connections because they don’t articulate with other bones, but why not model muscles as nodes, e.g. Diogo et al. 2015? (I’m not saying their approach was better, in fact yours makes a lot of intuitive sense, but when you deviate from previously published methods you should state your reasons.)

Validity of the findings

The conclusions are also more concise, and the links to the research questions are now clear. I think it would be even stronger if conclusion #2 were deleted (throughout the ms). Since embedded and glide sesamoids have very different network properties, it doesn’t really make sense to discuss them as a group, and I’m not even sure what it means. This is even more obvious when you get to conclusion #5, which states that embedded sesamoids are similar to canonical skeletal elements. You have a very clear story: embedded sesamoids have certain properties that are similar in some ways to canonical elements, whereas glide sesamoids have properties that are very different from either of the other two.

Additional comments

This is a very interesting study that raises important questions about the evolution and development of sesamoids and their role in the musculoskeletal network. In this revision, the authors have addressed many points raised by me and the other reviewers. I have some remaining minor questions and requests for clarification, which mainly center around the rationale for the study and the interpretation of results.

---

## Round 0.3 · accepted · Accept

I have checked the revisions and they are quite satisfactory. Thank you for your efforts, and congratulations!